# The Role and Mechanism of MicroRNA 21 in Osteogenesis: An Update

**DOI:** 10.3390/ijms241411330

**Published:** 2023-07-11

**Authors:** Revatyambigai Subramaniam, Ubashini Vijakumaran, Lohashenpahan Shanmuganantha, Jia-Xian Law, Ekram Alias, Min-Hwei Ng

**Affiliations:** 1Centre for Tissue Engineering and Regenerative Medicine, Faculty of Medicine, Universiti Kebangsaan Malaysia, Cheras 56000, Malaysia; p106228@siswa.ukm.edu.my (R.S.);; 2Department of Biochemistry, Faculty of Medicine, Universiti Kebangsaan Malaysia, Cheras 56000, Malaysia; ekram.alias@ppukm.ukm.edu.my

**Keywords:** *microRNA 21*, osteogenesis, bone resorption, bone homeostasis, exogenous *miR21*

## Abstract

MicroRNAs are short, single-stranded ribonucleic acids expressed endogenously in the body to regulate gene expression at the post-translational level, with exogenous microRNA offering an attractive approach to therapy. Among the myriad microRNA candidates involved in controlling bone homeostasis and remodeling, *microRNA 21* (*miR21*) is the most abundant. This paper discusses the studies conducted on the role and mechanism of human *miR21* (*hsa-miR21*) in the regulation of bones and the various pathways mediated by *miR21*, and explores the feasibility of employing exogenous *miR21* as a strategy for promoting osteogenesis. From the literature review, it was clear that *miR21* plays a dual role in bone metabolism by regulating both bone formation and bone resorption. There is substantial evidence to date from both in vitro and in vivo studies that exogenous *miR21* can successfully accelerate new bone synthesis in the context of bone loss due to injury or osteoporosis. This supports the exploration of applications of exogenous *miR21* in bone regenerative therapy in the future.

## 1. Introduction

Bone is a type of mineralized connective tissue that constantly undergoes remodeling to maintain the structural integrity of the skeletal system (i.e., bone homeostasis) and to adapt to mechanical stress or changes in the body’s needs throughout life by replacing damaged or old bone. Bone remodeling primarily consists of three phases, i.e., resorption, reversal, and bone formation [1,2], through a complex interplay of many factors. It primarily involves the interaction of two distinct bone cells, i.e., osteoblasts and osteoclasts.

### 1.1. Bone Remodeling and Homeostasis

Bone remodeling is an adaptation process of bone to external stimuli and the environment, and it begins with the bone resorption phase. Bone resorption is a process of mineralized bone removal via osteoclasts, and is guided by the receptor activator of the nuclear kappa-B ligand (RANKL), the receptor activator nuclear factor kappa-B (RANK), and osteoprotegerin (OPG). RANKL and OPG are primarily secreted by stromal cells, osteoblasts, and osteocytes, while RANK is expressed on the surfaces of osteoclasts and their precursors. The binding of RANKL to RANK activates NF-KB, which in turn upregulates c-Fos and NFATc1 in a series of processes that induce the differentiation of osteoblasts into mature osteoclasts [3]. Osteoclasts are multinucleated cells derived from a mononuclear lineage, and are compacted with Golgi complexes, mitochondria, and transport vesicles of lysosomal enzymes. Osteoclasts attach themselves to the bone, then release acid phosphatase and cathepsin K to break down the bone by proteolysis and acidification of the bone matrix and HA [4]. OPG competes with RANK for RANKL to avoid over-excessive resorption by inducing apoptosis of osteoclasts. Thus, the RANK/OPG ratio is crucial to determine the rate of bone resorption.

In the subsequent phase, known as reversal, osteoclasts reabsorb the bone surface for the purpose of new bone formation. Termination of the osteoclasts leaves the remaining collagen fragments exposed. The bone lining cell removes the fragments and forms a thin layer of new bone matrix to distinguish the old from the new [5].

This is then followed by the bone formation phase, or osteogenesis, which involves the (i) differentiation and (ii) maturation of osteoblasts (osteoblastogenesis), the (iii) synthesis of the bone matrix, its (iv) mineralization, and the eventual (v) formation of mature bone tissue. Osteoblastogenesis begins with the differentiation of mesenchymal stem cells (MSCs) residing in the periosteum or migrating from the surrounding tissues, such as the bone marrow into osteoprogenitor cells, also known as pre-osteoblasts, in response to triggers such as growth factors (e.g., BMPs, TGF-beta, IGF, FGF), cytokines, signaling proteins (e.g., Wnt, Notch, Shh proteins), and hormones (parathyroid hormone). Osteoprogenitor cells then undergo further differentiation into osteoblasts under the influence of specific signaling molecules and transcription factors, including Runx2 (Runt-related transcription factor 2) and Osterix, which drive the expression of genes involved in bone formation. Osteoblasts then synthesize and secrete the unmineralized organic matrix of bone, also known as osteoid, which mainly consists of collagen type I fibers, osteopontin, and osteocalcin. The osteoid provides the framework for mineralization and acts as a scaffold for the deposition of calcium and other minerals. The mineralization of the osteoid begins with the osteoblast forming specialized membrane-bound vesicles called matrix vesicles within their cytoplasm containing enzymes such as alkaline phosphatase, ions such as calcium and phosphate, and other molecules necessary for mineralization. Small amorphous mineral clusters serving as nucleation sites for the formation of hydroxyapatite crystals are released into the extra cellular space where they continue to grow, align and become integrated with the collagen fibers of the osteoid. Calcium and phosphate ions from the bloodstream are deposited onto the collagen scaffold, forming hydroxyapatite crystals. Mineralization by osteoblasts can be achieved either via intramembranous ossification or endochondral ossification mechanisms. In intramembranous ossification, bone is directly formed by mesenchymal stem cells differentiating into osteoblasts, which is followed by secretion of the bone matrix (osteoid) and mineralization. Endochondral ossification, which is more common, is a process by which mesenchymal stem cells first differentiate into chondroblasts, thus secreting a cartilaginous matrix, and then invasion by osteoblasts replaces the cartilage with the mineralized bone matrix [4,5]. As the bone matrix becomes mineralized, some of the osteoblasts become embedded within the matrix and differentiate into osteocytes. Osteocytes are the most abundant cells in mature bone tissue and play crucial roles in both maintaining bone health and responding to mechanical stimuli.

The delicate balance between bone resorption by osteoclasts and bone formation by osteoblasts is essential to ensure that the skeletal system remains strong and healthy. This balance is maintained by a complex interplay of hormones, growth factors, and cellular signaling pathways that modulate the activity of osteoclasts and osteoblasts. Various interactions of osteoclasts and osteoblasts have been studied to date, and most are orchestrated by the RANKL/RANK/OPG and Wnt signaling pathways [3]. Aside from signaling pathways, endocrine secretion hormones also contribute to bone remodeling by coupling osteoclastogenesis and osteoblastogenesis. Examples include growth hormones, insulin growth factors, glucocorticoids, sex hormones (estrogen and androgen), growth factors, prostaglandins, and cytokines [2].

### 1.2. The Coupling Mechanism between Osteoblasts and Osteoclasts

The coupling mechanism between osteoblasts and osteoclasts is crucial for maintaining bone remodeling homeostasis by balancing bone resorption and formation. This process is regulated by several mediators, including EFNB2-EPHB4, FAS-FASL, and NRP1-SEMA3A. During bone resorption, osteoblasts secrete TGF-β and IGF-1, which induce osteoblastic activity. Osteoblasts also secrete M-CSF, RANKL, and WNT5A, which promote osteoclastic formation [6]. EFNB2, expressed on the osteoclast cell surface, forms a bond with EFNB4 on the osteoblast surface to mediate bidirectional signal transduction between the two cells. EPBH2-mediated EPHB4 activation promotes osteoblastogenesis while EPHB4-induced EFNB2 activation interrupts C-Fos/NFATC signaling pathway, and thus working in reverse fashion from osteoblast to osteoclast to reduce osteoclast activity [7]. FASL is secreted in response to a paracrine signal to decrease osteoclast activity, whereas osteoblasts secrete FAS to increase osteoclast activity [8]. SEMA3A, produced by the osteoblast cell lineage, inhibits RANKL-induced osteoclast reactions and promotes osteoblast activity [9]. M-CSF, secreted by osteoblasts, is an important factor for cell proliferation, binding to C-FMS on the surface of osteoclasts to maintain the coupling mechanism. RANKL, highly expressed in osteoblasts, binds to RANK to initiate osteoclast differentiation, and OPG negatively regulates this process by inhibiting osteoclastogenesis by binding to RANKL [10]. WNT is also essential in bone remodeling, and WNT5A expressed in the osteoblast cell lineage binds to ROR2 on the osteoclast surface. WNT5A enhances bone resorption through the MAPK pathway [11].

### 1.3. Role of MicroRNA in Bone Homeostasis

MicroRNAs (miRNAs) are single-stranded non-coding RNAs consisting of 19 to 24 nucleotides. They modulate gene activity by binding to or degrading the target messenger RNA (mRNA), hence inhibiting the translation of mRNA into protein. As a result, the expression of specific proteins is suppressed while the upregulation of target proteins associated with the inhibited proteins is simultaneously orchestrated. This indirect mechanism enables miRNAs to facilitate the upregulation of certain proteins by inhibiting the expression of their negative regulators. This interplay leads to negative regulation of gene expression. As a result, specific gene expressions are downregulated or upregulated. In short, microRNAs play an essential role in cell proliferation and differentiation, apoptosis, the metabolism of fat, and resistance to stress [12].

MicroRNA plays a prominent role in both osteoblast and osteoclast differentiation by regulating bone formation through multiple pathways, involving a cascade of signaling pathways [12]. miR20 has been shown to upregulate the osteogenesis of human bone mesenchymal cells (hBMSCs) by downregulating peroxisome proliferator-activated receptor- (PPARỳ) and BMP activin membrane-bound inhibitor (BAMBI) signaling. According to the results, osteoblast markers BMP2, BMP4, Runx2, Osx, OCN, and OPN were elevated [12]. A different study showed that the downregulation of miR-133 and miR-135 inversely upregulates the Runx2 and Smad 5 osteogenesis gene regulators. miR-133 and miR-135 can bind to the 3′ untranslated region (UTR) of CTGF mRNA, resulting in the downregulation of connective tissue growth factor (CTGF) expression [13]. The downregulation of CTGF by miR-133 and miR-135 can potentially affect the balance between bone formation and bone resorption, leading to an influence on osteogenic differentiation and bone mineralization [14]. The action of miR 346 on T cell factor/lymphoid enhancer factor (TCF/LEF), a transcription factor of the Wnt/Catenin pathway, should also be noted. The Wnt/B-catenin pathway has been shown to enhance the activity of ALP in undifferentiated BMSCs to promote osteogenesis [15,16]. Downregulation of miR-31 was shown to suppress RANKL and induce the formation of osteoclasts to induce bone resorption [12]. Another example is that miR15b positively regulates osteoblasts by targeting Smurf1 to express Runx2 expression. miR 15b is a specific miR expressed in osteoblasts which binds to the mRNA of Smurf1 through complementary base pairing, specifically recognizing the 3′ untranslated region (UTR) of Smurf1 mRNA. This binding leads to the downregulation of Smurf1 expression. Smurf1 is an E3 ubiquitin that targets Runx2, the master regulator of osteoblast differentiation, and degrades it [17]. In terms of the expression of Runx2, when miR-15b targets and downregulates the expression of Smurf1, it indirectly leads to an increase in the expression of Runx2. Both positive and negative coordination of osteogenesis are regulated by miR through transcription factors. MicroRNAs can exhibit a dual role in gene regulation. While they typically inhibit the expression of specific proteins, in some cases, this inhibition leads to the upregulation of other proteins, resulting in a positive regulatory effect. Conversely, when microRNAs inhibit the expression of certain proteins and consequently alter or diminish their effects, this is referred to as negative regulation of the mechanism. These findings suggest that exogenous microRNA supplementation may be a possible therapeutic approach to overcome bone-related disorders.

## 2. Role of MicroRNA 21 in Bone Homeostasis

The discovery of microRNA in *Caenorhabditis elegans* (*C. elegan*) and humans, along with the study of their regulatory functions, explained gene expressions and genomics as an entirely new concept [18]. The discovery of miRNAs and their regulatory functions has shed light on the complexity and fine-tuning of gene expression. Previously, gene regulation was primarily attributed to transcription factors and other DNA-binding proteins. *Lin-4* was discovered in *C. elegans* as a short non-coding microRNA that regulates gene expression [1]. *Let-7* was the second microRNA discovered, and this was followed by many novel microRNAs that were identified in flies, worms, vertebrates, and plants [19]. These findings have led to remarkable progress toward the study of diverse microRNAs. *miR21* was one of the earliest to be identified and studied due to its role in health and diseases [20]. Essentially, *miR21* regulates cell growth, migration, and invasion, and is also expressed in immune modulator cells, B and T cells, monocytes, macrophages, and dendritic cells (DCs) [21]. In general, *miR21* suppresses the target mRNA’s gene of interest by binding to its 3′ UTRs. This binding leads to the degradation of the mRNA and inhibits its translation. This cascade of reactions can result in the upregulation or downregulation of specific gene expressions, which, in turn, has positive or negative effects on osteogenic differentiation and mineralization. *miR21* plays a role in promoting osteogenesis by safeguarding Runx2, which regulates the synthesis of other proteins related to bone formation. Additionally, *miR21* serves as a crucial regulator for inducing RunX2 [22].

Recent updates have stated that miR21 may act with either pro-inflammatory or anti-inflammatory effects in any healthy or pathological environment. This condition depends on the microenvironment; complex signaling pathways; signaling radiated by immune cells; and extracellular signals such as 12-O-Tetradecanoylphorbol-13-acetate (TPA)/Phorbol 12 myristate 13 acetate (PMA), lipopolysaccharides (LPS), interleukin 6 (IL6), tumor growth factor (TGF)/bone morphogenetic proteins (BMP), and many more. The end results of these signals stimulate the role of *miR21* as a negative or positive regulator of an inflammation environment at the transcriptional or post-transcriptional level [23]. It has been labeled as a cancer-promoting microRNA, or “oncomiR”, and has since been a target for diagnostic or prognostic markers and therapeutic candidates (anti-microRNA therapy) [24]. However, it is still not clearly understood. This miR21 is released as a biomarker of tumor formation, as it is widely transported in the exosomes. In addition, miR21 itself elicits an inflammation reaction to escalate tumor progression or to orchestrate a general immune response. Despite this association, there is also a rising body of evidence that *miR21* plays an integral role in osteogenesis, and, thereby, there have been attempts to develop microRNA therapy for treatment of bone loss [25]. Table 1 provides evidence of the role of microRNA in bone formation in both in vitro and in vivo studies. This review attempts to provide an update on the studies conducted in the last five years on the role and mechanism of human miR21 (hsa-*miR21*) in osteogenesis, and to evaluate its value in exogenous microRNA therapy for the promotion of bone regeneration. The present findings provide support for the fundamental process of bone formation, known as osteogenesis, which is enhanced by both endogenous and exogenous *miR21*. These results indicate a significant advancement towards utilizing miR for bone regeneration in the context of current applications involving carriers and therapies.

## 3. Empirical Evidence of MicroRNA-21 in Regulating Bone Homeostasis

### 3.1. In Vitro Studies

In vitro studies play a crucial role in establishing preliminary data and identifying potential outcomes, which can then be further investigated in vivo. Both in vitro and in vivo, bone marrow-derived mesenchymal stem cells (MSCs) were commonly used, and bone-related markers were assessed through techniques such as PCR and Western blotting. Mineralization was determined through staining protocols by following the introduction of *miR21* exogenously, and comparing between treated and untreated groups. The majority of studies utilized MSCs extracted from mice, specifically the Sprague–Dawley and C57BL/6 strains, with some using wild-type or *miR21* knock-out mice. Additionally, Yang extracted MSCs from Labrador dogs and treated them with LacZ-transfected *miR21* mimics and inhibitors [30].

Most of the studies reviewed in this paper used in vitro models to establish their findings, and only a few extended their investigations to animal models for in vivo studies. These studies utilized mesenchymal stem cells (MSCs) to assess bone-related markers through PCR and Western blotting, and staining protocols were employed to study and measure mineralization upon the exogenous introduction of *miR21*. They utilized bone marrow-derived MSCs extracted from mice, such as the Sprague–Dawley (rat strain) and C57BL/6 mice, which were either wild-type or *miR21* knock-out. In addition, Yang extracted BMSCs from Labrador dogs and treated them with LacZ-transfected with *miR21* mimics and *miR21* inhibitors [30]. The studies also utilized various cell lines, such as the macrophage cell line RA W264.7, the pre-osteoblastic mouse cell line MC3T3, the osteoblast-like MG63, and the precursor cell line 4B12. RA W264.7 is an osteoclast cell line derived from BALB mice transformed with the Abelson leukemia virus, which Wang used to study the coupling mechanism of *miR21* in both bone formation and bone resorption [28].

Smieszek transfected the MC3T3 cell line, derived from Mus Musculus mice, to investigate the effect of *miR21* on inducing osteoblast proliferation [32]. The MG63 cell line was incorporated with the Ti-SrHA-21 scaffold by Geng to study the effect of *miR21* on osteogenesis [29]. The 4B12 cell line, derived from the young calvaria of a mouse, is primarily used to analyze the differentiation of TRAP-positive multinucleated cells into osteoclasts. Smieszek utilized this cell line to compare the effects of *miR21* on both bone formation and bone sorption. Besides using MSCs and cell lines from mice, CD4+ T cells extracted from the mouse spleen were also employed to measure the effect of *miR21* on promoting osteogenesis [32]. Wu concluded that T cells play a role in the osteoclast mechanism, and found that *miR21* promoted the secretion of RANKL by activated T cells, leading to the promotion of osteoclast activity and thereby increasing osteoblast activity [29]. Xu analyzed the regulation of the *miR21*/STAT3 signal on odontoblast differentiation of dental pulp stem cells (DPSCs) in an inflammation microenvironment constructed by TNF-a [26]. The study assessed the role of *miR21* in orthodontic bone development by measuring the dentin matrix acidic phosphoprotein (DMP1) and dental sialophosphoprotein (DSPP) proteins.

Overall, most in vitro studies were conducted to investigate the effect of *miR21*, along with its scaffold, directly on cell lines or primary cells from animals to demonstrate that *miR21* positively regulates osteogenic differentiation and mineralization by promoting the expression of key osteogenic factors, such as ALP, RUNX2, OPN, and OSX. The rates of bone formation and bone resorption were measured by RT-PCR, Western blot, and ELISA analysis. Immunostaining, ALP, and alizarin staining were carried out to detect the presence of minerals.

### 3.2. In Vivo Studies

In vivo studies provide more significant outcomes that are more relevant to future human subjects. The mineralization rate and bone formation or healing rate were measured using imaging techniques, RT PCR, and Western blot analysis. Various staining methods are used to determine the rates of bone healing and mineralization. Improved movement ability of animal subjects after treatment or surgery indicated successful bone growth and healing in some studies. Wang and Li demonstrated the novel contribution of *miR21* by comparing its effects in wild-type mice with those in knock-out mice [22,28]. Endogenous *miR21* expression in wild-type mice BMSCs after osteoinduction was analyzed, revealing that these cells naturally express *miR21*. This study also compared the effect of endogenous *miR21* expression with that of exogenously introduced *miR21*. Wild-type mice treated with *miR21* mimic showed enhanced new bone formation due to *miR21* overexpression. Li also conducted an in vivo study to evaluate *miR21′*s osteogenic properties [28].

In this particular investigation, rapid maxillary expansion was induced in both the knock-out and wild-type mice, and the expression of *miR21* in BMSCs and the duration of new bone formation were evaluated. The knock-out mice exhibited slower migration of periosteal cells, implying the significance of *miR21* in new bone formation. To further confirm this, exogenous *miR21* was administered to the knock-out mice, which showed a substantial difference in the rate of new bone formation. Geng implanted four Titanium (Ti)-coated with SrHA and *miR21* rods in the hind legs of thirty New Zealand rabbits. X-ray and CT scan analyses revealed that all samples had excellent osseointegration, particularly Ti-SrHA-21, which exhibited significant osteoconductivity and osteoinductivity [29]. Osteoconductivity relates to the physical support provided by a material for bone growth, while osteoinductivity refers to the biological signaling ability of a material to promote the differentiation of cells into bone-forming cells. Both properties are crucial in the field of bone regeneration and tissue engineering, and different materials may possess varying degrees of osteoconductivity and osteoinductivity. In another study by Yang, the osteogenic impact of lentivirus transfected with *miR21* and integrated with β-TCP scaffold was examined in a canine model [30]. Histological examination demonstrated that more new bone was formed during the Lenti-*miR21*/β-TCP BMSCs scaffold implantation. These discoveries represent a breakthrough in terms of proving the osteogenic effect of *miR21* in new bone formation, as well as its enhanced healing capacity. To evaluate the efficacy of scaffolds combined with *miR21*, further clinical trials in humans are needed.

## 4. Pathways Regulated by *miR-21* in Bone Homeostasis

### 4.1. Smad Pathway

Smad is an intracellular signaling protein molecule composed of Smad 1–9 members. These molecules react by phosphorylating transforming growth factor-beta (TGF β) or bone morphogenic protein (BMP) [31]. Transforming growth factor-beta (TGF-β) and bone morphogenic proteins (BMPs) BMP 2, BMP 4, and BMP 7 are crucial in osteoblast differentiation. Both TGF β and BMP have significant roles in bone development. BMP binds to the type 1 receptor and activates the Smad pathway by phosphorylating Smad 1/5/8, which then forms complexes with Smad 4. This complex is then translocated into the nucleus and acts on transcription factors such as RunX2 and Osx. RunX2 plays a crucial role in regulating MSCs to differentiate in osteoblast lineage; it is known as the master gene in regulating osteoblast cell formations. Osx is involved in osteogenesis by inducing bone matrix formation and initiating the differentiation of MSCs into an osteogenic lineage [31]. RunX2 and Osx induce pre-osteoblast cells to differentiate into osteoblasts [33]. Additionally, *miR21* modulates the expression of target genes involved in bone-forming cells’ osteoblast proliferation and differentiation. *miR21* regulates the Smad signaling pathway by targeting and suppressing the expression of Smad 7, a negative regulator of the osteogenesis pathway. Thus, this leads to the activation of the Smad 1/5/8 pathway to control bone formation.

In X. Li et al. discovered that *miR21* is involved in the osteogenic differentiation of BMSCs via the Smad7-Smad1/5/8 and RunX2 pathways [16]. Both in vitro and in vivo experiments were conducted on wild-type and knock-out *miR21* mice. The knock-out mice without *miR21* exhibited lower bone formation than the wild-type mice, supporting the idea that endogenous *miR21* plays a role in bone formation. Through transfecting BMSCs from knock-out mice with siRNA to remove Smad 7, this study found that *miR21* directly targets Smad 7, leading to increased expression of Rux2 and ALP, as seen Western blot and PCR analysis. Overall, this paper provides solid evidence for the role of *miR21* in the Smad1/5/8 pathway [16].

The synergistic effect of strontium-substituted hydroxyapatite and *miR21* in improving bone remodeling and osseointegration was demonstrated by Geng through both in vitro and in vivo studies [29]. In the study, Ti-SrHA (titanium-strontium hydroxyapatite) loaded with *miR21* was implanted in rabbits. The Ti-SrHA-*miR21* provided better surface adhesion, promoting increased cell proliferation compared to the non-coated implant. The nanostructured and hydrophilic properties of this Ti-SrHA combination allowed for uniform distribution of the nanocapsules of *miR21*. Endogenously added *miR21* increased the osteoblast cell proliferation and the ALP expression compared to the non-treated group. The imaging analyses using SEM, MicroCT, and X-ray projected accelerated mineralization in Ti-SrHA-*miR21* compared to Ti, Ti*-miR21*, and Ti-SrHa. The participation of SrHa increased the osteoblast proliferation rate as well, but was not as efficient as the participation of *miR21*. This proved that Ti-SrHA-*miR21* led to rapid bone healing. The paper concluded that COL-I, RUNX2, OPN, OPG, and OCN, which are osteogenesis-related genes, were significantly upregulated as a synergic effect of Ti-SrHA, along with *miR21*. This finding suggests the probable participation of *miR21* in the Smad pathway, as was demonstrated in a previous study by Li [16,29].

Wang studied the role of *miR21* in the reconstruction of maxillary bone defects through the Smad pathway [28]. In this study, two groups of mice, i.e., the wild-type and *miR21* knock-out mice, were compared to assess the rate of bone formation in maxillary bone defects. The author postulated that *miR21* promoted the osteogenesis of BMSCs via the Smad7-Smad1/5/8-Runx2 pathway, based on the previous work by Li [28]. Maxillary transverse deficiency (MTD), a skeletal deformity of the craniofacial region, was created in wild-type and *miR21* knock-out mice. The healing of the palatal suture was monitored to evaluate the role of *miR21* in bone formation. The histochemical analysis and micro-CT scanning proved the role of *miR21* in bone healing and new bone formation, with the results comparatively explaining the presence and absence of *miR21* function, thus proving that *miR21* plays a role in osteogenesis. This was further confirmed by an analysis of ALP and OCN gene regulation, which was upregulated in the wild-type mice. The authors proposed possible pathways, such as the P13K/BMP9/Smad signaling pathway/ß catenin pathway, based on previous studies by Meng and Liu [16,21]. The focus of this paper was solely to prove the role of *miR21* in bone healing by measuring the mineralization rate and the rate of healing [28].

Sun et al. studied *miR21* in nanocapsules and its role in promoting the early bone repair of osteoporotic fractures by stimulating the osteogenic differentiation of bone marrow mesenchymal stem cells. The main objective of this study was to investigate the role of *miR21* in osteogenesis and the efficiency of the nanocapsule scaffold to deliver the micro-RNA. It is crucial to deliver the degraded micro-RNA directly into the cell’s nucleus. In this study, an osteoporotic bone defect model was generated in Sprague–Dawley rats, which were then injected with the carrier O-carboxymethyl chitosan (CMCS)/*miR21* composites. The zeta potential yielded satisfactory results for the *miR21* in encapsulation into nanocapsules. The confocal image showed that the *miR21* FAM-tagged nanocapsule was successfully taken up by the BMSCs. CMCS was synthesized in favor of and highly sensitive to the metalloproteinase secreted by fractured regions. Thus, this eased the uptake of the fractured region and increased the potential to deliver microRNA into the cells successfully. The bone formation was analyzed with alizarin staining, and it was identified that CMCS/mimic miR showed more new bone formation than the negative control. CMCS, with a negative control score, increased ALP and RunX2 after a longer treatment period. This was due to the self-renewal capacity of the BMSCs cell by itself, and the CMSA was excluded as a potential autoinducer. The author concluded that the injection of nanoencapsulated CMCS/*miR21* can effectively treat osteoporotic conditions due to the role of *miR21* in osteogenesis and the efficiency of CMCS as an effective delivery tool [15].

### 4.2. RANKL/RANK/OPG Pathway

Nuclear factor -k β (NF-k β) ligand (RANKL), tumor necrosis factor (TNF), and macrophage colony-stimulating factor (M-CSF) are osteoclast-stimulating factors [34]. RANKL is a transmembrane protein found as a membrane-bound, secreted protein resulting from proteolytic cleavage, and is expressed by synovial cells or secreted by activated T cells. RANKL interacts with RANK further by activating TRAF6, leading to cascade’s mitogen-activated protein kinase (MAPKs), ERK, p38, JNK (c-Jun N terminal kinase), and AKT (protein kinase B). T cell-produced RANKL stimulates osteoclast formation through c-Fos. In contrast, osteoprotegerin (OPG) is a decoy receptor that binds to RANKL and inhibits osteoclastogenesis through the nuclear factor of activated T cells (NFATc1), NFATc1 stimulates OSX, an essential transcription factor for osteoblasts [33].

Smieszek’s paper investigated MC3T3 pre-osteoblast cells, MC3T3 *miR21*-inhibited cells and the 4BI2 pre-osteoclast precursor cell line. MC3T3/4B12, the merge between pre-osteoclast and osteoblast cell lines, resulted in increased tartrate-resistant acid phosphatase (TRAP), matrix metalloproteinase (MMP9), and Cathepsin K (Ctsk) genes, which are actively involved in osteoclastogenesis. MC3T3*_inh_*_21_ cells reduced Coll-1, OCL, OPN, and RunX2, the key regulators of osteogenesis. The inhibition of miR21 upregulates the osteoclast-suppressor-programmed cell death protein 4 (PDCD4), and, vice versa, downregulates osteoclasts. Thus, this explains the significance of *miR21* in osteogenesis. The coupling role of (OPN) in osteoblast differentiation facilitates the attachment of osteoclasts at the resorbed matrix region, which is emphasized in this study. OPN plays a crucial role in the early stage of bone remodeling by differentiating osteoblast cells. During paracrine interaction between osteoblast cells, OPN stimulates the attachment of osteoclasts and the release of RANKL [2]. In this study, the expression of OPN was directly proportional to *miR21*, as seen in MC3T3*_inh_*_21_ cells. The expression of OPN was reduced in MC3T3inh21 cells and increased in MC3T3/4B12 cells. Western blot analysis further narrowed down the findings according to the molecular weights. OPN protein, in the range of 35kDA, was expressed at an increasing trend in MC3T3/4B12 cells, and decreased in the MC3T3*_inh_*_21_/4B12 group. With no significance, expression was observed in MC3T3*_inh_*_21_ in the range of 66kDA. This relates to the different isomers of OPN in the different mechanisms of bone remodeling, with a dual role. MC3T3*_inh_*_21_ showed a decreased OPG and was accompanied by an increased RANKL; thus, this ratio indicated osteoblast activity. OPG played an inhibitory role by blocking the interaction between RANKL and RANK. Interestingly, the MC3T*_inh_*_21_/4B12 co-culture was designed in this study to understand the dual role of miR21. MC3T/4B12 was leading towards osteoclastogenesis, as expected, since both the osteoblast and osteoclast precursor cells were present with miR21 in both of the cell lines. On the other hand, MC3T*_inh_*_21_/4B12 showed that the ratio of RANKL/OPG was decreased, thus indicating that the absence of miR21 affects osteoclastogenesis as well. These findings firmly establish the dual role of *miR21* in osteoblast and osteoclast coupling through RANKL and OPN [32].

Wu et al. designed a study to investigate the effects of *miR21* on orthodontic tooth movement via the RANKL/OPG balance in T cells. The orthodontic tooth movement (OTM) model was established in C57BL/6 wild-type (WT) and *miR21* knock-out mice (*miR21*KO). For the other group, T cells from the wild-type mice were injected into the *miR21* KO mice two days before the study. Micro-CT scan results showed that the WT mice had larger OTM distances compared to the *miR21*KO mice injected with T cells, and the complete *miR21*KO mice expressed the shortest OTM distance. An immunohistochemical analysis was carried out to measure the OTM distance. The results were similar to the micro-CT scan, where a greater distance was seen in WT followed by *miR21*KO with T cells, and the shortest distances was observed in *miR21*KO mice. The osteoclast (OC) from the dental root with exerted pressure was then counted using a light microscope, and the same pattern was observed, whereby the highest score of osteoclasts was found in the WT followed by the T cells injected with *miR21* KO, and the lowest was found in the *miR21*KO group. The knock-out mice exhibited retarded tooth movement distances compared to the wild-type. This could be due to the dysfunction of the osteoclast in the absence of *miR21*. The blood serum of both wild-type and knock-out mice was analyzed to examine the influence of *miR21* on T cells’ OPG/RANKL/RANK expression levels. The results showed that RANKL expression in the WT type was higher compared to the *miR21*KO mice. Further, T cells were isolated from both WT and *miR21*KO mice, then cultured and harvested for RANKL analysis. This analysis proved that *miR21* targets T cells to regulate OPG/RANKL/RANK expression. The expression of T cells with overexpression of *miR21* was also analyzed through rt-PCR. Both T cells from the wild-type and the knock-out mice were analyzed, and this resulted in reduced secretion of RANKL in the knock-out mice due to the absence of *miR21* [31]. We concluded that miR21 regulates osteoclastogenesis through RANKL and T cells secreted in the inflammation microenvironment, and also actively stimulates RANKL to perform osteoclastogenesis. Tooth movement was increased in *miR21* accompanied by CD4+T cells compared to in miR21 by itself. It can be postulated that *miR21,* under various types of microenvironment stimulation, balances the OPG/RANKL differently. In this study, the pre-existing inflammation condition switched the role of miR21 from a bone formation to a bone resorption stimulator.

### 4.3. STAT3 Pathway

Bone healing is initiated by an inflammatory mechanism, followed by bone formation processes. This inflammatory reaction is initiated by lymphocytes; monocytes; neutrophils; macrophages; and secretions such as IL1-IL6, TNF, and many more. Janus kinase (JAK)/signal transducer and activator of transcription (STAT) are primarily involved in cell metabolism and differentiation in inflamed microenvironments. JAKs belongs to the protein tyrosine kinase (PTK) family, and its primary role is to act as a STAT [35]. JAK induces the phosphorylation of STAT, which then migrates to the nucleus and binds to target genes to induce MSCs for osteoblast differentiation [31]. In bone homeostasis, IL-6 is produced by osteoblast cells to promote bone resorption by osteoclasts. IL-6 may negatively regulate osteogenesis through SHP2/MEK/ERK and SHP2/p13K/Akt2, or may positively regulate osteogenesis through the JAK/STAT3 pathway [36]. JAK/STAT3 is a pathway activated by a series of cytokinins, and it can also be regulated by *miR21* by suppressing PTEN/PDCD4. Vice versa, JAK/STAT3 activation can also induce the expression of miR21 in an inflammation microenvironment [37].

Dental pulp stem cells (DPSCs) are a type of mesenchymal stem cell that can differentiate into osteogenesis lineages by stimulating *miR21* and STAT proteins. In this study, *miR21* and STAT 3 proteins are activated by a concentration of tumor necrosis factor (TNF-α) as low as 10ng/mL. As a result of this activation, odontoblast differentiation is induced, resulting in the upregulation of dentin sialophosphoprotein (DSPP) and dentin matrix acidic phosphoprotein 1 (DMP1) mineralization-related genes. Cell suspensions of dental pulp were extracted from normal human-impacted third molars and treated with 1, 10, 50, and 100 ng/mL concentrations of TNF-α. The 10 ng/mL treatment provided a suitable microenvironment for odontoblast differentiation. It was proven through ALP and Western blot analysis that this resulted in increased expression of DSSP, DMP1, and *miR21*. This was further investigated by inhibiting STAT3 with cucurbitacin I (Cuc I), which resulted in the downregulation of *miR21* expression. This was proven by a chromatin immunoprecipitation study in which chromatin precipitated with STAT3 antibodies was significantly enriched for the *miR21* promoter sequence. This study concluded that TNF-a activated STAT3 and *miR21* were upregulated, and the Smad pathway was activated for bone formation, resulting in odontoblast differentiation of DPSCs [26]. The activated STAT3 increased osteogenesis through BMP2, resulting in the upregulation of the osteogenesis genes ALP, DSSP, and DMPI. STAT3 is involved in the regulatory network of embryonic stem cells (ESCs), and participates in the LIF and BMP signaling pathways, which play essential roles in self-renewal, reprogramming, and pluripotency in ESCs [11]. STAT3 can be phosphorylated by the stimulating inflammatory factor IL-6, an inflammatory factor [26]. Inflammation initiates a cascade reaction by secreting pro-inflammatory factors such as IL-1, IL-6, and TNF-α, which also play roles in differentiating BMSCs. IL6 positively regulates JAK/STAT3 signaling pathways [36].

### 4.4. PTEN-PI3K/Akt Signaling Pathway

Phosphatase and tensin homolog (PTEN) play crucial roles in bone formation by regulating the phosphoinositide 3-kinases/protein Kinase B (P13/Akt) pathway. PTEN is a bilipid protein phosphatase targeted to phosphatidylinositol-3,4,5-triphosphate (PIP3), a product of phosphatidylinositol 3 kinases (P13K). The inhibition of PTEN, as well as P13K, activates the downstream activators pyruvate dehydrogenase kinase 1(PDK1), AKT/PKN, and G proteins Rac1/cdc42, leading to cell growth and proliferation [38]. PTEN is well known for its function in osteogenesis through the AKT/P13K pathway. Interestingly, miR21 has the potential to regulate PTEN to modulate both osteoblastogenesis and osteoclastogenesis.

In a study by Yang, the suppression of PTEN-activated P13/AKT signaling, which, as a result of increased osteogenesis and overexpression of *miR21*, increased the BMSC capacity for bone regeneration [30]. The authors hypothesized that *miR21* enhanced osteogenesis through the PTEN pathway. To prove this, BMSCs were transfected with *miR21* mimic and *miR21* antagonist, then tested in vitro and analyzed through Western blot. The authors showed that the *miR21* mimic increased hypoxic-inducible factor 1 alpha (HIF-1α), P-AKT, BMP2, OPN, OCN, and P13K, and downregulated PTEN expression. The mechanism of the PTEN pathway was further demonstrated when the expression level of PTEN increased as the inhibitor of P13K was introduced to the mimic *miR21* through the introduction of lenti-miRNA-21 + LY294002 (PI3K inhibitor). Therefore, this paper proved the role of *miR21* in osteogenesis through the PTEN/PI3K/Akt/HIF-1α pathway [30]. As a proof of concept, a lenti-miRNA-21/β-TCP/BMSC scaffold was implanted to treat rat calvarial bone defects and canine mandibular defects. Both the rat and canine bone defects showed increased new bone formation, thus signifying the role of *miR21* in osteointegration. Contradicting Yang’s findings, Wang also investigated the role of *miR21* through PTEN on macrophages where tartrate-resistant acid phosphatase (TRAP) and *miR21* levels were upregulated in RANKL-induced RA W2647 cells [28,30]. They concluded that *miR21* upregulated osteoclastogenesis and promoted bone resorption through the P13K/AKT signaling pathway by targeting PTEN in RA W264.7 cells (macrophage cell line). The investigation was carried out in vitro by transfecting RA W264.7 (murine macrophage cell line) with miR21 mimic, *miR21* inhibitor, and *miR21* mimic transfected with PTEN inhibitor. These groups were treated on bovine bone slices, and bone resorption was measured through TRAP staining. It was found that PTEN and *miR21* were working in opposite directions; when one expression was suppressed, the other was overexpressed. *miR21* mimic with the P13K inhibitor resulted in the reduced expression of PTEN compared to the *miR21* mimic. The authors stated that this was due to the role of *miR21* in osteoclastogenesis through the P13K/AKT pathway. They concluded that *miR21* directly suppresses PTEN and, as a result, activates P13k/AKT/NH-KBs/NFATc1, which initiates an osteoclastogenic mechanism. However, while the P13K inhibitor inhibited the P13K/AKT pathway, osteoclastogenesis may have been an effect of the RANKL expression derived from the RA W264.7. Furthermore, there was no further research conducted to prove that osteoclastogenesis was a direct effect of the PTEN through the P13K/AKT pathway. The upregulation of *miR21* in macrophages during osteoclastogenesis explains its role in bone resorption and osteoclast differentiation. In the future, authors should also include osteogenic markers to concretely prove that osteoclastogenesis is the only mechanism of action. Macrophage colony-stimulating factors (MCSF) and RANKL receptors induced the nuclear factor of activated T cells 1 (NFATc1), NK-NF-κB, and c-Fos to initiate osteoclastogenesis [28]. Interestingly, both Yang et al., 2019 [30], and Wang et al., 2020 [28], inhibited PTEN to regulate osteoblastogenesis and osteoclastogenesis, respectively, through the same P13K/AKT pathway. Wang’s study utilized a macrophage cell line, which was induced towards inflammation, while Yang’s study utilized BMSCs, bone progenitor cells that favor bone formation. It may be postulated that, as part of bone homeostasis, *miR21* arising from an inflamed microenvironment along with RANKL secretion from the macrophages may be a form of feedback inducing bone progenitor cells to promote osteogenesis.

### 4.5. WNT/β- Catenin Pathway

P13K also activates AKT, which enhances pGSK3b and plays a role through the canonical WNT signaling pathway. Canonical WNT/β- catenin binds to LPS 5/6 (low-density lipoprotein receptor-related protein). Followed by the binding of WNT receptors to frizzled (FZD), this forms a ternary complex resulting in the phosphorylation of WNT. This was followed by the assembly of disheveled (Dlv), which induces the phosphorylation of Lrp5/6. This resulted in the inhibition of the AXIN complex protein, which is composed of glycogen synthase kinase 3 beta (GSK-3β), adenomatous polyposis coli (APC), and casein kinase I (CK1). AXIN is responsible for destructing β-catenin. Thus, Dvl blocks the phosphorylation of Axin by (GSK-3β), which is an essential factor for the phosphorylation of β- catenin. This then causes the accumulation of β-catenin in the cytoplasm, eventually migrating into the nucleus and regulating the gene expression of RunX 2, thus increasing the formation of osteoblast precursor cells [39]. Besides this, *miR21* also promotes the phosphorylation of the glycogen synthase kinase three beta (GSK-3β), which accumulates beta-catenin in the cytoplasm, then targets TCF3, a gene which enhances osteogenesis [21]. The mentioned pathways (Smad, RANKL/RANK/OPG, STAT, and beta-catenin) are interconnected, and play important roles in bone homeostasis. The Smad pathway is involved in bone formation and is regulated by *miR-21*. In this pathway, *miR-21* targets and suppresses the expression of Smad 7, a negative regulator of osteogenesis. By inhibiting Smad 7, *miR-21* promotes the activation of the Smad 1/5/8 pathway, which enhances bone formation. The RANKL/RANK/OPG pathway is crucial for regulating the balance between bone resorption (osteoclast activity) and bone formation (osteoblast activity). *miR21* influences this pathway by regulating the expression of key factors*. miR21* can upregulate the expression of RANKL, a protein that stimulates osteoclast formation, and can downregulate the expression of OPG, a protein that inhibits osteoclast formation. This imbalance between RANKL and OPG promotes osteoclast activity, leading to increased bone resorption. The STAT pathway is involved in various cellular processes, including bone metabolism. *miR21* is known to play a role in this pathway. By targeting specific genes or regulators within the STAT pathway, *miR21* can influence the signaling and transcriptional activity of STAT proteins, which may impact bone homeostasis. The β-catenin pathway is a critical signaling pathway involved in bone development, maintenance, and regeneration. It regulates the differentiation and activity of osteoblasts, which are responsible for bone formation. MiR-21 has been found to interact with this pathway, although the exact mechanisms are still being investigated. It may regulate the expression of certain genes or modulate the activity of key proteins within the beta-catenin pathway, impacting bone remodeling processes. In summary, miR-21 regulates multiple pathways involved in bone homeostasis, including the Smad, RANKL/RANK/OPG, STAT, and β- catenin pathways. These pathways interact and collectively contribute to the regulation of osteoblastogenesis, osteoclastogenesis, and bone remodeling. *miR21′*s modulation of these pathways can have significant effects on bone health and may play a role in conditions such as osteoporosis or bone-related disorders (Figure 1).

## 5. *miR21* in Therapeutic Applications

Pharmaceutical companies are actively exploring alternative therapeutic molecules as substitutes for existing chemically composed drugs. These molecules need to fulfill the medical requirements in terms of pharmacokinetic availability, properties, safety, and efficacy. Presently, numerous miRNA molecules are undergoing clinical trials, and there is a significant body of literature, consisting of around 600 published articles, focusing on miRNA-based therapeutics. The first miRNA molecule to enter clinical trials was Miravirsen, which is currently in phase II trials across multiple countries [7]. This review paper examines several studies with promising potential for inclusion in the therapeutic approach. These studies aim to assess the role of exogenous *miR21*, with or without a carrier, in new bone formation, potentially enhancing the effectiveness of the current therapeutic approaches for bone healing and formation. Sun’s study consistently highlights the role of nanocapsulated *miR21* in the healing of osteoporotic fractures, demonstrating its potential to accelerate bone healing in osteoporotic patients. Yang also acknowledged the promising therapeutic effect of *miR21* on the facilitation of rapid bone formation. In Yang’s research, *miR21* was encapsulated with chitosan and administered via injection in gel form to an osteoporotic model. The efficient release of *miR21* stimulated bone repair in the osteoporotic model. Yang conducted experiments on canine and rat bone defect models to investigate the influence of *miR21* on new bone formation, suggesting that *miR21* regulates the PTE/PI3K pathway for osteogenesis. The study validated these effects through both in vivo and in vitro assessments. In the canine model, an osteoperiosteal defect model was introduced and treated with *miR21* incorporated into β-tricalcium phosphate (β-TCP), resulting in a remarkable increase in bone formation compared to the non-treated group. This further emphasizes the positive impact of *miR21* on bone healing. Geng’s research focused on studying the effects of titanium coated with strontium-substituted hydroxyapatite, incorporating *miR21* into bone healing using a rabbit model. The findings indicated that a combination of SrHA, *miR21*, and titanium promoted bone mineralization and strength. The consistent evidence of mineralization leading to new bone formation resulting from the addition of endogenous miR21 suggests its broad potential for implementation in therapeutic approaches for the treatment of bone-related disorders [15,29,30]. For efficient exogenous *miR21* delivery, most studies resorted to the use of carriers. To facilitate the efficient delivery of *miR21*, researchers have explored the use of carriers. These carriers can be diverse in nature, ranging from synthetic nanoparticles or liposomes to cellular carriers. Synthetic nanoparticles are engineered particles typically made of biocompatible materials such as lipids, polymers, or metals. They can be designed to encapsulate *miR21* within their structure, protecting it from degradation and enabling controlled release at the target site. These nanoparticles can be functionalized with specific ligands or antibodies to enhance their targeting ability towards specific cells or tissues. Liposomes are another type of carrier commonly used for miRNA delivery. They are spherical vesicles composed of lipid bilayers and can encapsulate *miR21* within their aqueous core. Liposomes are biocompatible and can be modified to improve stability, cellular uptake, and targeted delivery. In addition to synthetic carriers, cellular carriers have also been investigated for *miR21* delivery. These carriers are often derived from cells themselves, such as stem cells or immune cells. They can be engineered to produce and release *miR21* directly at the target site, exploiting the natural homing and tissue-penetrating abilities of the cells. The choice of carriers depends on various factors, including the desired target tissue, the stability of *miR21* during delivery, the desired release kinetics, and the potential side effects. Researchers continuously explore and optimize different carrier systems to achieve efficient and safe *miR21* delivery in the human system for therapeutic purposes [40,41]. The subject of the efficiency of microRNA delivery will require another extensive review.

## 6. *miR-21* in the Coupling Mechanism

The role of *miR21* in bone remodeling is determined by the inducing factor, which determines whether it will promote bone formation or bone reformation. Notably, *miR21* is involved in both bone resorption and bone formation mechanisms through its coupling role in osteoclastogenesis and osteoblastogenesis, respectively [42].

*miR21* is induced towards osteoclastogenesis by RANKL secretion. In cases of inflammation, IL6 can result in the overexpression of *miR21*, which activates STAT3 and inhibits OPG production, ultimately promoting RANKL gene activation. Conversely, inhibiting *miR21* would hinder STAT3 activation, thereby disrupting the OPG/RANKL pathway. The contribution of *miR21* to the coupling mechanism is entirely reliant on the inducing factor. The maintenance of bone remodeling homeostasis depends on various hormones, factors, and signaling molecules. RANKL binds to RANK, leading to the activation of TRAFs and a cascade of ERK, p38, JNK, and P13K [43]. This process triggers c-Fos, upregulates the expression of *miR21* gene, and downregulates PDCD4 protein levels. PDCD4 is a tumor suppressor that is involved in cell proliferation and progression. As a result of RANKL binding, NFATc1 and BMM are expressed to initiate osteoclast differentiation. NFATc1 acts as a co-factor with AP-1, composed of Fos/Jun proteins, to bind to osteoclast-specific markers such as TRAP and cathepsin. PDCD4 also affects the transcription factor AP-1, which regulates cellular differentiation, proliferation, and apoptosis. Studies show that *miR21* has binding sites for transcription factors such as Ap-1 and PU.1. PU.1 is a lineage-specific transcription factor that regulates various cell lineages, including osteoclasts. OPN, which plays a critical role in osteoclastogenesis, is regulated by PU.1. Transcription factors such as c-Fos and PU.1 increase *miR21* expression through Ap-1 and upregulate OPN, leading to a shift in bone homeostasis to bone resorption [44]. However, when OPN is upregulated, it can also promote osteogenesis by regulating the activity of osteoblasts. OPN can stimulate the differentiation and activity of osteoblasts, leading to new bone formation. Additionally, OPN can enhance the mineralization of the bone matrix by binding to hydroxyapatite, which is a key component of bone tissue. Therefore, the upregulation of OPN can promote the switch from bone resorption to bone formation, thus increasing bone mineralization. Therefore, *miR21* may have specific expressions and functions at different stages of bone resorption. The phosphorylation of Smad proteins is involved in the regulation of bone morphogenetic proteins (BMPs), which allows Smad molecules to enter the nucleus and modulate gene expression to initiate osteogenesis. BMP 4 and TGFβ can increase the transcription of *miR21*, leading to inhibition of Smad 7. This allows BMPs to bind to the type 1 receptor, which activates the Smad pathway through the phosphorylation of Smad 1/5/8. These molecules then form complexes with Smad 4 and translocate to the nucleus to act on transcription factors such as RunX2 to induce osteogenesis. Conversely, BMP6 inhibits *miR21* through the AP-1 binding site. The expression of *miR21* is selective and may be induced at different stages to support the progression of differentiation. Previous studies have suggested that *miR21* has a dual role in balancing bone resorption, as it can either accelerate or inhibit it. Transcriptional and post-transcriptional regulation are involved in maintaining *miR21* expression, which is dependent on several selective factors. BMP 4 and TGFβ induce *miR21* expression and are responsible for osteogenesis.

In contrast, *miR21* is induced for osteoclastogenesis by RANKL secretion. In cases of inflammation, IL6 can result in the overexpression of *miR21*, which activates STAT3 and inhibits OPG production, ultimately promoting RANKL gene activation. Conversely, inhibiting *miR21* would hinder STAT3 activation, thereby disrupting the OPG/RANKL pathway. The contribution of *miR21* to the coupling mechanism is entirely reliant on the inducing factor. The maintenance of bone remodeling homeostasis depends on various hormones, factors, and signaling molecules. The role of *miR21* in bone remodeling is determined by the inducing factor, determining whether it promotes bone formation or bone resorption (Figure 2).

## 7. Conclusions

This review paper presents a body of evidence that demonstrates the involvement of *miR21* in osteogenesis, which promotes the synthesis of new bone. It also shows that *miR21* plays a role in bone regeneration through multiple pathways and has a dual function in bone formation, including both osteoclastogenesis and osteblastogenesis. The coupling mechanism of *miR21* enables it to regulate both bone resorption and bone formation, and maintaining a balance is critical for ensuring healthy bone regeneration. This article delves into the intricate details of various signaling pathways that are controlled by *miR21* during bone remodeling, such as Smad, Wnt, Catenin, PTEN, STAT3, RANKL, and Sprouty. Furthermore, this review highlights the potential of *miR21* as an adjuvant therapy that can be administered directly to patients or in combination with implants to promote osteointegration.

## 8. Knowledge Gap and Future Studies

Additional research is needed in order to explore the potential outcomes that could arise from introducing exogenous *miR21* at various stages of healing, especially in the presence of inflammatory or pathological conditions like osteoporosis. The role of *miR21* as an oncomiR warrants further investigation to assess its impact as both a regulator of osteogenesis and a promoter of cancer. In the process of osteoclastogenesis, the RANKL/RANK signaling pathway is activated, leading to the activation of various regulatory molecules such as P13K, JNK, ERK, and p38, as well as the transcription factor c-Fos. This activation of c-Fos is thought to contribute to the upregulation of *miR21* expression The increased levels of *miR21* are associated with the reduction in PDCD4, a tumor suppressor protein that initiates the development of metastatic cells. As a result, this could potentially contribute to the classification of *miR21* as an oncomiR. Nonetheless, in the majority of instances, elevated levels of *miR21* are detected in various types of cancer, and its expression is often higher in advanced malignancies. Epigenetic modifications could have a substantial influence on *miR21* expression, with overexpression potentially leading to the development of carcinogenic cascades. On the other hand, the preexisting carcinogenic microenvironment resulting from the rapid differentiation of malignant cells may upregulate *miR21* expression via various pathways, thereby making it the most highly expressed miRNA in carcinogenic conditions. This could explain why *miR21* is considered as a diagnostic marker for oncomiRs (Feng and Tsao 2016). Despite this, it remains unclear whether *miR21* causes cancer or whether changes in its expression levels are associated with the progression of cancer. Overall, the effects of exogenous *miR21* administration on osteogenesis or carcinogenesis are context-dependent, and may vary depending on the target cells and other factors such as the presence of other miRNAs or signaling pathways. Long-term preclinical studies are needed in order to fully understand the potential therapeutic applications and risks associated with exogenous *miR21* administration. Future studies should be conducted to investigate the impact of exogenous *miR21* on bone remodeling and its role as an oncomiR in tumor-bearing animal models.

## Figures and Tables

**Figure 1 ijms-24-11330-f001:**
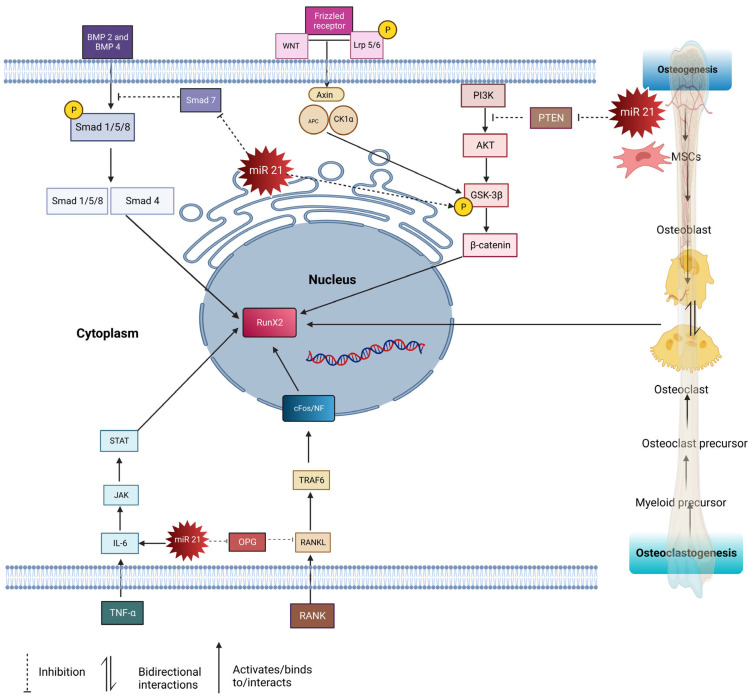
The pathways involved in osteogenesis and osteoclastogenesis regulated by *miR21*: RANKL/OPG, PTEN/AKT/P13K, BMP2/SMAD 1/5/8, TNFα/IL-6/JAK/STAT3, WNT/βCatenin. PTEN/AKT/P13K pathway: *miR21* suppresses PTEN, thus activating the PI3K-AKT-GSK3β pathway, promoting β-catenin in the nucleus to modulate the expression of osteogenesis-related genes. TNFα/IL-6/JAK/STAT3 pathway: IL 6 activates *miR21*, resulting in the STAT pathway being upregulated and increasing osteogenesis. RANKL/OPG pathway: RANK ligand binds to RANKL receptors and activates TRAF6 cascades, leading to the activation of osteoclasts. *miR21* inhibits the secretion of OPG, and the imbalance ratio of OPG/RANKL upregulates osteoclastogenesis. WNT/βCatenin pathway: WNT binds to Fz and Lrp5/6, causing Dvl to phosphorylate Lrp5/6, leading to the accumulation of β-catenin, which then translocates to the nucleus to induce osteogenesis. miR21 phosphorylation upregulates GSK-3β, thus increasing the accumulation of β-catenin. Smad pathway: BMPs bind to type 1 receptor and activate the Smad pathway by phosphorylating Smad 1/5/8, forming complexes with Smad 4. This complex translocates into the nucleus and acts on transcription factors such as RunX 2 to induce osteogenesis. miR21 inhibits Smad 7, thus upregulating the Smad 1/5/8 pathways.

**Figure 2 ijms-24-11330-f002:**
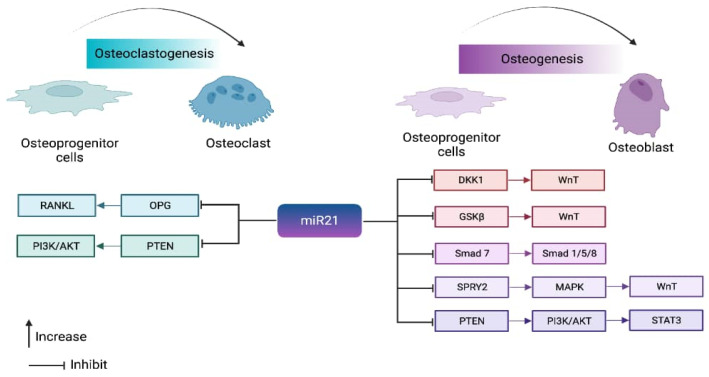
The dual role of microRNA 21 in both osteoclastogenesis and osteoblastogenesis. Osteoslatogenesis: As a result of *miR21* inhibition of OPG, the RANKL pathway is activated. Similarly, the inhibition of PTEN by miR21 leads to the activation of the P13K/AKT pathway. Osteogenesis: The inhibition of DKK1, GSKβ, Smad 7, SPYR2, and PTEN by miR21 results in the activation of the WnT and Smad pathways. Interestingly, inhibition of PTEN can also result in osteogenesis by activating the Stat3 pathway. The Figure 2 illustrates the dual role of *miR21* in osteoclastogenesis and osteogenesis through the same PTEN pathway highlights the complexity and context-dependent of *miR21* regulation in bone formation.

**Table 1 ijms-24-11330-t001:** Evidence of the role of microRNA-21 in osteogenesis in both in vitro and in vivo studies.

Reference	Aim	In Vitro Study	In Vivo Study	Carrier	Findings	Signaling Molecules/ Pathways	Conclusion
Li et al., 2017 [16]	To investigate the effects of *miR21* on the osteogenic differentiation of BMMSCs and the signaling pathways that influence the biological behaviors of BMMSCs	BMMSCs from the bone cavities of wild-type (WT) and microRNA-21 knock-out (*miR21*-KO) mice. These BMSCs were osteoinduced and assessed for *miR21* expression. BMSCs from WT mice were transfected with exogenous miR21 mimic. All cells were then analyzed through RT-PCR, alizarin and ALP staining, and Western blot for miR21 expression.	5-week-old female C57BL/6J wild and knock-out mice. *n* = 5 (20–22 g). The calvarial bone defect model was bilaterally established in the WT and *miR21*-KO mice. The defective bones were analyzed after 2 months using a stereomicroscope and H&E staining. BMMSCs of WT were transfected with *miR21,* and BMMSCs of *miR21*-KO mice were transfected with Smad7 siRNA.		1. RT-PCR and Western blot analyses showed that the expression of key transcription factors that regulate osteogenesis were significantly upregulated in the WT mice BMSCs compared to *miR21*-KO mice BMSCs. 2. BMMSCs of WT mice transfected with exogeneous *miR21* resulted in enhanced mineralized nodule formation and increased osteogenic marker expressions. BMMSCs from *miR21*-KO mice showed much lower ALP activity compared to those from WT mice. 3. Western blot showed increased expression of ALP, Runx2, and Smad1/5/8 expression in BMMSCs of *miR21*-KO mice transfected with Smad7 siRNA.	*miR21* directly inhibits Smad7, upregulates the Smad1/5/8-Runx2 pathway, and promotes osteogenesis.	*miR21* is indispensable in the osteogenic differentiation of BMMSCs, and *miR21* regulates osteoblastic bone via the Smad7-Smad1/5/8-Runx2 pathway.
Xu et al., 2018 [26]	To analyze the regulation of *miR21*/STAT3 and odontoblast differentiation of DPSCs in the inflammatory microenvironment induced by TNF-alpha.	Human-impacted third molars collected from patients (18–28 years of age). *n* = 15. DPSC cells from 3rd passage were treated in a TNF-alpha induced inflammatory microenvironment (1–100 ng/mL). Cells were also transfected with STAT3 siRNA, control siRNA, and miRNA inhibitor as (−) control. MTT assay, ALP and alizarin staining, Western blot, and RT- PCR were carried out.	NIL		1. The expressions of (dentin matrix acidic phosphoprotein) DMP1 and (dental sialophosphoprotein) DSPP, stimulated by TNF-alpha, at low concentrations (10 ng/mL). 2. Increased expression of STAT3 and *miR21* was observed in low concentrations of TNF-alpha. 3. PCR showed increased DMP1, DSPP, miR21, and STAT3 in the nucleus due to stimulation by low concentrations of TNF-alpha. 4. Chromatin was precipitated with STAT3 antibodies and was significantly enriched for the *miR21* promoter sequence. 5. Expressions of DMP1 and DSPP significantly decreased when transfected with siSTAT3 or treated with Cucurbitacin I (Cuc I). Besides that, pri miR21 also increased under this condition. This shows STAT3 is directly involved in the expression of *miR21* during odontoblast differentiation.	In the TNF-alpha-induced microenvironment, STAT3 protein became activated (p-STAT3) and increased bone morphogenetic protein receptor 2 (BMPR2) expression by stimulating the maturation of *miR21*.	STAT3 was activated by TNF-alpha, stimulated *miR21* maturation in DPSCs, and mediated odontoblast differentiation.
M. Li et al., 2020 [27]	To elucidate the effects of *miR-21* on the mid-palatal suture-induced bone remodeling by expanding the palatal suture.	BMSCs from *miR-21*−/− and WT mice were cultured Cell migration and cytotoxicity assays were carried out	Wild-type and KO C57BL/6 mice. 6 weeks old, male. 19–21 g. They were used to establish animal models of rapid maxillary expansion (RME). 4 groups were established: (i) WT (control); (ii) WT + RME; (iii*) miR21*-/1; (iv) *miR21*−/− + RME. The maxillae were harvested by euthanizing 3 mice every 1, 3, 7, 14, and 28 days. (PCR) genotyping for WT and *miR21*−/− mice was performed. *miR-21*−/− mice with expansion force received agomir-21, and controls were treated with PBS (0.2 mL) through intraperitoneal injection every 3 days for 2 consecutive weeks Histology, histochemical, and immunohistochemical studies were conducted.		1. Histological analysis showed new bone formation in WT mice to be more rapid than in the *miR-21*−/− mice. 2. Alizarin and Calcein complex markers showed stronger expression in WT mice compared to *miR21*−/− mice 3. Expression of ALP and OCN revealed increased expression in WT mice compared to miR-21−/− mice 4. TRAP staining indicated increased RANKl and decreased OPG expression in miR21−/− (RME) mice. 5. Increased osteoprotegerin (Opg) expression and decreased Rankl expression in *miR21*−/− mice were observed using immunohistochemical staining and immunofluorescence. 6. *miR21*−/− mice received agomir-21, and PCR showed increased expression of ALP, OCN, and OPG to RANK ratio.	miR-21 affected bone remodeling by coupling reaction through RANKL/OPG. *miR-21* regulated the proliferation and migration of BMSCs and bone formation during RME in WT mice.	Findings showed that overexpression of *miR21* can promote osteogenic differentiation and accelerate fracture healing.
Wang et al., 2020 [28]	To investigate the effects of *miR21* on osteoclastogenesis and bone resorption, as well as the potential molecular mechanisms underlying the Pten-PI3K/Akt signaling pathway in osteoclasts.	The murine macrophage cell line RA W264.7 was cultured with RANKL to induce osteoclastogenesis. RA W264.7 cells were transfected with: miR NC; *miR21* mimic; *miR21* inhibitor; *miR21* mimic + LY29400 (P13K inhibitor). Trap staining, Western Blot, and RT PCR were carried out. Bone resorption assay: Bovine bone slices were treated with the transfected cells/controls, and the bone resorption was calculated as the pit area/total bone area of each slice.			1. TRAP analysis indicated that *miR21* was progressively upregulated during osteoclastogenesis in vitro. 2. RT-PCR Western blot analyses showed decreased Pten protein expression in the *miR21* mimic group, thus suggesting that miR21 negatively regulated PTEN. A high-level miR21 was detected during osteoclastogenesis. 3. The percentage of bone resorption was significantly higher in the *miR21* mimic group compared to NC and inhibitor. 4. The *miR21* mimic group decreased the protein expression levels of Pten, but upregulated p-A kt and NFATc1 expression. 5. The bone resorption assay resulted as significantly higher in the mimic group compared to the other groups.	PTEN regulates osteoclastogenesis in the RANKL-induced RA W264.7 osteoclast cell line by activating the P13K/Akt signaling pathway.	*miR 21* was upregulated in osteoclastogenesis and promoted bone resorption by activating the PI3K/Akt signaling pathway via targeting PTEN.
Geng et al., 2018 [29]	To evaluate biodegradable SrHA and *miR21* composite coatings on Ti implants, and the effects on bone formation and osseointegration by promoting osteogenesis and inhibiting osteoclastic activity.	MG63 osteoblast-like cell line was used Cell morphology, density, cell proliferation, ALP activity, and cell distribution were studied	Mature New Zealand white rabbits. *n* = 30. 2.5–3 kg. Rabbits were implanted with four of the Tis coated with SrHA and *miR21* rods at the distal femur and tibial plateau of the hind legs	*miR21* was encapsulated in acrylamide polymer. Ti-SrHA was synthesized *miR21* was coated on Ti-SrHA Encapsulated *miR21* was observed under TEM Ti-SrHA-21 was observed through confocal imaging and X-ray photoelectron spectroscopy (XPS)	1. The *miR21* nanocapsules, synthesized by an in situ polymerization method, showed a spherical structure with a uniform diameter of ~30 nm. 2. Confocal images suggested that the *miR21* nanocapsules were distributed evenly on the surfaces of Ti-21 and Ti-SrHA-21. 3. H&E and Masson Trichome staining results indicated that SrHA and *miR21* had synergistic effects on bone remodeling and new bone mineralization. 4. Fluorescence microscopy images of the MG63 cells incorporated with Ti-SrHA-21 exhibited the best cell-spreading property and highest cell density compared to other groups. 5. Ti-SrHA-21 significantly increased the CD31 expression in the early stages of surgery. CD31 is an important endothelial marker that could contribute to the development of new blood vessels and, thus, plays an important role in bone formation and increased osteogenesis-related gene expression (including COL-I, RUNX2, OCN, and OPN). 6. The X-ray and CT scan showed that all samples exhibited good osseointegration, especially Ti-SrHA-21, which showed remarkable osteoconductivity and osteoinductivity. 7. Raman spectra results indicated that the degree of new bone mineralization increased with the healing time.	*miR21* promotes stem cells’ survival and osteogenesis, presumably via the Smad7-Smad1/5/8-Runx2 and Akt pathways.	SrHA and *miR21* synergistically promote angiogenesis, osteogenesis, anti-osteoclastic, osseointegration, bone mineralization, and bone–implant bonding strength.
Yang et al., 2019 [30]	To investigate the role of *miR21* in promoting osteogenesis in bone marrow-derived stem cells (BMSCs) in vitro.	BMMSCs from Labrador dogs (approximately 2 years old). BMSCs were transfected with *miR21* mimic, *miR21* inhibitor, or Lenti-*miR21*(LacZ) using oligofectamine.	Rat. *n* = 18. A calvarial bone-defect rat model was created. Lenti-*miR21*/β-TCP/BMSC scaffolds in a canine mandibular defect model.	Carrier Lenti-*miR21*(LacZ), *miR21* mimic, and inhibitor-transfected BMSCs were seeded into β-TCP scaffolds (200 μL per scaffold, at a density of 1.0 × 10^7^ cells/mL)	1. RT-PCR for BMSCs infected with Lenti-*miR21* showed that BMP-2, Runx2, OCN, and OPN mRNA levels increased. 2. Western blotting revealed that HIF-1α, Akt, and PI3K protein levels increased nearly twofold after *miR21* mimic transfection, and, in contrast PTEN decreased. 3. Histological examination showed that more new bone formed in the Lenti-*miR21*/β-TCP BMSCs scaffold implantation group compared with the other two scaffolds (Lenti-LacZ β-TCP and Lenti-LacZ β-TCP BMSC) 4. Micro-CT images revealed new calvarial bone formation in the Lenti *miR21*/β-TCP/BMSC scaffold group. 5. Gieson’s staining showed that bone formation was observed in the Lenti-*miR21*/β-TCP/BMSC group, and less in the Lenti-LacZ/β-TCP/BMSC group. 6. Bone formation and mineralization were assessed histomorphometrically via tetracycline, calcein, and ARS fluorescence. Data indicated that Lenti-*miR21* enhanced ossification in the induced BMSCs, and effectively promoted new bone formation.	*miR21* overexpression increased BMSC migration and HIF-1α activity via the PTEN/PI3K/Akt pathway.	miRNA-21 overexpression in BMSCs may offer great therapeutic promise for rapid bone formation during the healing process by modulation of the osteogenesis process via the PTEN/P13K pathway.
Wu et al., 2019 [31]	To investigate the effects of microRNA-21 (*miR21*) on orthodontic tooth movement (OTM).	CD4+ T cells were purified from the spleenocytes of WT or *miR21*−/− mice. CD4+ T cells from WT mice were transplanted into *miR21*−/− mice. ELISA, Western blot, and RT-PCR analyses were carried out.	WT C57BL and *miR21*−/− mice. Male. 20–25 g 8 weeks old *n* = 6 An orthodontic tooth movement model was established in C57BL mice. The mice were sacrificed on the 14th day after this procedure, after which the maxillae were removed and the blood was collected by retro-orbital bleeding. Micro-computed tomography was used to measure the distance of tooth movement. Histological, histomorphometric, and immunohistochemical analyses were carried out.		1. Micro-CT scan-indicated OTM distance of *miR21*−/− mice was markedly less than that of WT mice. 2. H&E staining showed that the width of the periodontal ligament on the pressure side of the WT was lower than that in the *miR21*−/− mice. 3. The serum concentration of RANKL in *miR21*−/− mice was less compared to WT mice. 4. RT-PCR demonstrated that RANKL was downregulated in *miR-21*−/− mice, and WT mice induced the secretion of RANKL in activated T cells. 5. OTM distance revealed that the distance in *miR21*−/− mice with T cells was greater than that in *miR21*−/− mice without T cells, and less than that in WT mice. 6. RANKL/OPG ratio was significantly increased after the infusion of T cells in *miR21*−/− mice. 7. Macroscopic observation showed that the orthodontic tooth distance of miR- 21−/− mice with T cells was increased compared with that of *miR21*−/− mice without T cells. 8. The amount of OCs expression in *miR21*−/− infused with T cells was higher compared with that in WT mice, and the fewest OCs were seen in the *miR21*−/− mice.	*miR21* deficiency constitutively suppresses tooth movement due to reduced RANKL circulation, which leads to a reduced number of OCs *miR21* improves OCs via the *Pdcd4*/*Cf*os pathway, which influences the expression of the RANKL level secreted by T cells, enhances OC differentiation, and guides the osteoclastogenesis process.	*miR21* is closely related to CD4+ T cells in immune responses and stimulates the release of RANKL. As a result, it stimulates the differentiation of OCs.
Wang et al., 2020 [28]	To investigate the effect of *miR21* on bone reconstruction by inducing maxillary bone defects in wild-type (WT) and *miR21* knock-out (*miR21*-KO) mice, and exploring these mice as maxillary bone defect models.		WT and *miR21*-KO C57BL/6 mice. Female mice. 20–22 g. A round bone defect with a diameter of 1.2 mm was established at the maxillary bone on the palatal side of the incisors in each mouse. *miR21*-KO + agomir mice were injected with 100 μL of *miR21* agomir intra-peritoneally every three days. The WT and *miR21*-KO mice received 100 μL PBS injections. Bone histology, histomorphometry, and immunohistochemical analyses were carried out.		1. The lengths of the maxillary bone defects, as measured by H&E staining, were significantly longer in the *miR21*-KO group compared with the WT group. 2. Masson staining and the quantitative analyses also showed less new bone staining in the miR-21-KO group compared to WT mice. 3. The *miR21*-KO group exhibited lower ALP and OCN expressions than the WT group. 4. The value of BV/TV, as calculated by micro-CT, indicated that more new bone was detected in the *miR21*-KO + agomir group than in the *miR21*-KO group. 5. H&E staining showed that the length of the maxillary bone defect was decreased in the *miR21*-KO + agomir mice.	As concluded based on Li X et al., *miR21* was found to promote osteogenesis of BMSCs via the Smad7-Smad1/5/8-Runx2 pathway. The results obtained in this study show that *miR21* improved bone development. The vast differences can be seen in the miR KO and the miR KO + agomir.	*miR21* was crucial for the bone reconstruction of maxillary defects in vivo, which suggests that it could be a therapeutic target for maxillary bone defects.
Sun et al., 2020 [15]	To explore the effect of nanoencapsulation (*miR21*) on the healing process of osteoporotic fractures as a new alternative method to accelerate the healing of osteoporotic fractures.	BMSCs were isolated from osteoporotic ovariectomized (OVX) rats. Flow cytometry to detect ALP expression and calcium deposition in BMSCs isolated from osteoporotic ovariectomized (OVX) rats.	Sprague–Dawley (SD) rats Elderly; 14-month-old. A model of female osteoporotic bone defects was generated using Sprague–Dawley rats. *n* = 30. Groups: A: CMCS/n (*miR21*); B: CMCS/n (NC-miR); C: saline. 3 months after the procedure, the lumbar bone mineral density and trabecular bone microarchitecture of the lumbar were measured. Histological and immunohistochemical studies were conducted	- *miR21* or nanoencapsulated *miR21* (acrylamide polymer) and O-carboxymethyl chitosan (CMCS) were mixed until they formed a gel-like material CMCS/n (*miR21*) - DLS and TEM analyses were carried out	1. Electrophoresis and zeta potential analyses confirmed that the encapsulation efficacy of the nanocapsules for *miR21* was satisfactory. 2. The results of ALP staining, toluidine blue staining, and oil red O staining confirmed that BMSCs derived from the OVX rat model could be differentiated from osteoblasts. 3. The result of confocal microscopy showed that significant green fluorescence was present in the cytoplasm of BMSCs after incubation with n (*miR21*) labeled with FAM. 4. Alizarin red staining confirmed that BMSCs treated with n (*miR21*) showed increased mineralization. 5. Micro-CT showed that the defect site was significantly bridged with new cancellous bone in the CMCS/n (*miR21*) group compared with the CMCS/n (NC-miR) and control groups. Quantitative analysis of alizarin red staining showed that the *miR21*-treated group had higher calcium nodule formation than that of the n (NC-miR) group or the control group. 6. *miR21* promoted the expression of ALP and RUNX-2 genes in BMSCs isolated from OVX rats and inhibited the expression of caspase-3, thereby enabling the healing of osteoporotic bone defects.	*miR21* accelerates the healing of osteoporotic bone defects by promoting the early expression of the ALP and RUNX-2 genes in BMSCs.	*miR21* could promote early bone repair in osteoporotic bone defects by stimulating the osteogenic differentiation of BMSCs.
Smieszek et al., 2020 [32]	To investigate the impact of *miR21* inhibition on pre-osteoblastic cell differentiation and paracrine signaling towards pre-osteoclasts, using an indirect co-culture model of mouse pre-osteoblast (MC3T3) and pre-osteoclast (4B12) cell lines.	A pre-osteoblastic mouse cell line MC3T3 and an osteoclast precursor cell line 4B12 were used. MC3T3 cells were transfected with *miR21* inhibitor and then osteo-induced, followed by co-culturing with osteoclast precursor cell line 4B12. RT PCR, Western blot, SEM, ALP, and immunostaining were performed			1. MC3T3 transfected with *miR21* inhibitor and co-cultured with 4B12 showed a significant increase in Ocl. 2. Decreased mineralization was observed in cultures of MC3T3 with blocked activity of *miR21*. 3. Images obtained with a confocal microscope confirmed that OPN expression decreased after *miR21* inhibition. 4. Increased levels of transcripts for RANK, TRAP, CTSK, CAII, and MMP-9 in 4B12 co-cultured with MC3T3 without an inhibitor were correlated with increased pre-osteoclast activity. 5. Decreased expression of *miR21* in MC3T3 pre-osteoblasts cell line resulted in the loss of their bone-forming capability. 6. The inhibition of *miR21* expression attenuated the paracrine activity of preosteoblasts, which was associated with increased apoptosis of pre-osteoclasts in the co-culture model.	*miR21* regulates the differentiation of progenitor cells into bone-forming cells via the Smad7-Smad1/5/8-Runx2 pathway.	*MiR21* may have a dual role in the process of osteoblast–osteoclast coupling, probably because its targets (e.g., RANKL and OPN) are regulated dynamically during the process of osteogenesis.

## Data Availability

No new data was generated from this review.

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
