# Peer review of "The Role and Mechanism of MicroRNA 21 in Osteogenesis: An Update"

_ijms, 2023, doi:10.3390/ijms241411330_

Round 1
Reviewer 1 Report
This review article is focused on the role of MicroRNA 21 in osteogenesis. I have the following comments/suggestions for authors to improve the draft.
1. Section 2 – Role of MicroRNA 21 in Bone Homeostasis: Authors have focused on the introducing miR 21; while section says its role in bone homeostasis. Please consider re-writing this section in context to bone homeostasis.
2. Section 3.1 – Paragraph 2 (starting line 15) content is same as paragraph 1 (starting line 1). Looks like an oversight from the authors. Please make sure that content is not repeated anywhere in the manuscript.
3. Words “in vitro and in vivo” should be italicized throughout the manuscript.
4. Section 3.1 – line 10 – Sprague Dawley is a rat strain, not mice.
5. Section – 4 – Pathway Section – This section looks a dry run through the literature; authors should consider summarizing each pathway section at the end of respective sections.
6. Figure 1 quality should be improved.
No Comment! It looks fine!
Author Response
Dear Reviewer;
I am writing to express my sincere gratitude for your valuable comments and feedback on my recent review. Your insights and suggestions have proven to be immensely helpful in improving my work, and I wanted to take a moment to acknowledge and appreciate your efforts. Hereby I have attached my corrections.
Thank you
Please see the attachment

Reviewer 2 Report
The review article is comprehensive and I do not have any recommended changes.
Author Response
Dear reviewer;
I am writing to express my sincere gratitude for your valuable comments and feedback on my recent review. Your insights and suggestions have proven to be immensely helpful in improving my work, and I wanted to take a moment to acknowledge and appreciate your efforts. Hereby I have attached my corrections.
Thank you
Please see the attachment

Reviewer 3 Report
Dear Sirs,
although your work and reports valuable information I will suggest a moderate editing of the English language.
terms as sought, daughter cells, "to induce an accurate epigenetic modification to induce the osteogenic differentiation" should be considered and improved.
Author Response

(The authors gave the same response as above.)

Reviewer 4 Report
Dear Authors,
The manuscript with title “The role and mechanism of microRNA 21 in osteogenesis: An update” reviews the role and function of miR-21 in bone homeostasis pathways. As our understanding of bone homeostasis about how bone formation and bone degradation (or absorption) is coupled is still lacking, the content of this review, in which miR-21 is put forward as one of the factors that may couple these processes, is highly relevant. Also, in the context of RNA technologies that have been developed to edit genomes (e.g., CRISPR/Cas) and RNA technologies that are used in treatment of diseases (e.g., targeted treatments, vaccines), the content of this review is relevant for the scientific community. Moreover, bringing the individual work of many scientists together in an overview is useful and adds to our overall integrated understanding of in this case bone molecular regulation and physiology. The reviewer appreciates the in-depth details provided in the current manuscript.
Several comments and remarks can be found below.
A first comment is the usage and definitions of the terms ‘osteoclastogenesis’ and ‘osteogenesis’. Osteoclastogenesis implies the formation (differentiation) of the cells, i.e., the osteoclasts. Thus, this term does not relate to the active process that osteoclasts do, which is breaking down or resorbing bone. If indeed, the authors want to focus on the cell formation mechanism (it appears to be in section 1.2), that is from osteoclast precursor cell to functioning osteoclast, this should be clarified in the manuscript. Assuming cellular differentiation mechanisms are the focus of the manuscript, the opposite term should be ‘osteoblastogenesis’. In itself, osteogenesis implies a process of bone formation. However, a clear definition of what bone formation is, is missing from the manuscript, as the first lines (L24-25) imply that bone is mostly consistent of cells. Osteogenesis is a two-part process, with initially the formation of bone matrix, mostly consistent of collagen type I, and subsequently the mineralisation of the collagenous scaffold. These two processes, matrix formation and mineralisation should be clearly defined, and it should be specified about which of the two processes the authors write in context of the functions of miR-21. In the current manuscript osteogenesis is used for both processes, but a clear and detailed indication of which of the two processes miR-21 has an effect on would be a valuable addition to the manuscript. See the works of Witten et al. and Drabikova et al. about showing how matrix formation and mineralisation can be uncoupled. In conclusion, the authors need to clarify when they mean cell differentiation processes (osteoclastogenesis vs osteoblastogenesis) and when they mean bone formation (osteogenesis), with the latter further clarified into the matrix deposition part or the mineralisation part, or both.
The introduction would read more clearly if a bit more text would be provided on how bone is formed and which cells and cellular differentiation steps of these cells are involved in forming mature bone.
Often the terms ‘negative regulation’ and ‘positive regulation’ are used. However, these remain as vague descriptions and often leave the reader wondering what the authors exactly mean. Although effort has gone into greatly detailing pathways, the usage of positive and/or negative regulation makes statements or conclusions vague again. For example, in section 1.3 the authors state that miRNAs modulate gene activity by interfering with translation of mRNA, stating that this is negative regulation. In the next sentence, both up- and downregulation of expression is possible, was stated. The reviewer interprets this as potentially both positive or negative regulation, but again specification would be necessary to correctly understand what positive or negative regulation is. A second example, in L83- 84 miR-15b is stated to positively regulate osteoblasts by targeting SMURF1 to express RUNX2. What exactly is the positive regulation? The initiation of more pre-osteoblast differentiation, or the number of pre-osteoblasts increase, or something else? Thus, clearly stating with specific examples of what the authors mean by positive or negative regulation would help the readability of this manuscript. L86,
what is meant by ‘positive and negative’ coordination of osteogenesis’? The bone homeostasis balance between osteoclast and osteoblasts or the balance between bone formation and bone resorption?
L91-93: I do not agree that the discovery of miRNAs and the better understanding of their function through gene expression analyses and genomics (what do the authors mean with this term?) is such a new field. However, it is currently highly relevant.
L97-98: reference is made to miR-21 as being one of the earliest identified miRNAs. Why do the authors not reference that original work? Personally, I find it important that original articles and papers with first findings or descriptions are referred to. This remark brings me also to question why only the last five years are chosen to be reviewed? I think it is always necessary to review the entire time series of papers on a certain specific topic unless a clear reason for choosing a specific time window can be given. Older work is not less relevant, on the contrary.
L119-120: ‘microRNA therapy to promote bone regeneration’, although touched upon in this review, much more fundamental knowledge is given instead of specifically miRNA as possible therapeutic component. I miss a bit a discussion specifically towards this topic, where all fundamental knowledge gathered in this review is integrated.
In several locations throughout the manuscript, the authors contradict themselves. These contradictions should be resolved or clarified.
The first paragraph of section 3 is repeated, although in slightly different words, in the next paragraph.
In section 3 examples are given of in vitro and in vivo studies. Several times statements are made that measurements were taken from imaging techniques (e.g., x-ray, microCT, scans, staining techniques). Measurements can be taken from images or scans via software, but cannot be taken directly from visualisation techniques. Please clarify how measurements were taken from output of visualisation techniques. In addition, if more or new bone was visualised, could it be clarified if it was the matrix or the mineral component that was visualised, measured or noted upon?
Also in section 3, terms such as ‘osteoconductive’ and ‘osteoinductive’ have been used. Could the authors please clearly define these terms and the context in which they use such terms.
At the start of section 4, binding of TGF-Beta to its receptor should phosphorylate SMAD molecules and not vice versa? Also, the first lines are a repetition.
Section 5 L385-389: I do not know if I agree with the statement on ‘an overall increase in bone density and strength’. Although, that is what may ultimately happen, miR-21 may facilitate shifting the bone homeostasis balance from resorption to bone formation, in particular the mineralisation component, as collagen matrix needs to be mineralised before it can be resorbed and thus remodelled. Thus miR- 21 may act as a subtle regulator so that correct bone homeostasis and thus also bone resorption can happen, rather than a promotor for increased bone strength. Maybe an alternative route of thinking that could be explored by the authors. Thus, also on line 426, I personally think miR-21 is rather a bone homeostasis balancer or coupler of bone formation and resorption processes rather than a promotor of rapid bone formation.
A last major comment is about all the small details that relate to consistency and editing. Because of the use of large numbers of abbreviations and an inconsistent use of when abbreviations are written in full or clarified, the manuscript is very difficult to read. The first time an abbreviation is used, it should be written in full and if needed clarified. If this is in a table, the full-length term could be summed below the table or in a clearly referenced abbreviations list. Several mistakes in abbreviations
were made (LPS instead of LRP, P13K instead of PI3K as examples). Capital use mistakes in abbreviations were also present. In addition, an inconsistent use of abbreviations, with miR-21 as prime example (four to five different versions throughout the paper) and many others, makes this manuscript very hard to read. Also, inconsistency in nomenclature of genes (e.g., human should always be capital and italics) versus proteins (human should be capitals), inconsistent use of either the use of gene notation or use of knock-out (even KO was used), and inconsistency of how the table was filled (some studies included detailed techniques while others did not although there were results of techniques not indicated). The table columns are so closely apposed that text from one column almost runs into the next column, a better spacing should be provided. The reviewer was also wondering if the findings in the table should solely focus on mechanisms relevant to bone homeostasis, and if the conclusion column contains the conclusions of the referred papers or the conclusions made by the authors. There are also many instances where there are too many spaces or spaces are missing. The terms in vitro and in vivo should be consistently in Italics and also Mus musculus (species name never capital letter). There are several places where text or the final part of the sentence appears to be missing. A severe consistency and editing overhaul of the manuscript is necessary.
Finally, some work is necessary on the language in this paper.
See comments and suggestions for authors
Author Response

(The authors gave the same response as above.)

Round 2
Reviewer 1 Report
No further comment. Authors have all the concerns.
No comment; it looks good.
Reviewer 4 Report
Dear Authors,
The reviewer would like to thank the authors for their work and tackling the comments properly. The column spacing in the table remains very close, thus making it hard to read in the table. If methods or other essential information from papers that where included in the table are missing, would it not be opportune to not that in the table, so that the reader knows that information is not available? Finally, a very detailed proofread should identify the last inconsistencies considering spacing and different versions of micro-RNA abbreviations, and a few language inconsistencies.
Thank you again.
See comments and suggestions for authors.